# In Vitro Probiotic Characteristics and Whole Genome Sequence Analysis of *Lactobacillus* Strains Isolated from Cattle-Yak Milk

**DOI:** 10.3390/biology11010044

**Published:** 2021-12-29

**Authors:** Juanshan Zheng, Mei Du, Wei Jiang, Jianbo Zhang, Wenxiang Shen, Xiaoyu Ma, Zeyi Liang, Jiahao Shen, Xiaohu Wu, Xuezhi Ding

**Affiliations:** 1Key Laboratory of Yak Breeding Engineering, Lanzhou Institute of Husbandry and Pharmaceutical Sciences, Chinese Academy of Agricultural Sciences, Lanzhou 730050, China; sfazjs1228@sina.com (J.Z.); dumeicaas@126.com (M.D.); zhangjb9122@163.com (J.Z.); liangzeyi1@163.com (Z.L.); sjh18437913123@163.com (J.S.); 2Key Laboratory of Veterinary Pharmaceutical Development, Ministry of Agricultural and Rural Affairs, Lanzhou Institute of Husbandry and Pharmaceutical Sciences, Chinese Academy of Agricultural Sciences, Lanzhou 730050, China; gw18848970753@126.com (W.J.); shane095@foxmail.com (W.S.); 18903865563@163.com (X.M.)

**Keywords:** *Lactobacillus* strains probiotics, whole genome sequencing, functional characteristics

## Abstract

**Simple Summary:**

Cattle-yak milk is rich in numerous bioactive substances that function beyond their nutritive value. Given the high content of conjugated linoleic acid (CLA), this exponent suggests a potential role for cattle-yak milk in preventing cancer, heart disease, and metabolic disorders, such as diabetes. It is expected to isolate probiotic strains with natural and excellent characteristics in optimizing human health. However, little research data and much fewer studies on probiotics from cattle-yak milk are reported. Therefore, in the present study, the beneficial strains from cattle-yak milk were identified to evaluate probiotic characteristics of potential strains, and to screen potential probiotics with an emphasis on making better use of milk in such extreme climatic environments. These findings support the potential application of cattle-yak milk in human health, animal food processing, and drug production in the future as a potential candidate for probiotics.

**Abstract:**

Cattle-yak milk is an important raw material and an indispensable source of high-quality food for local farmers and herdsmen to produce ghee, milk residue, yogurt, and other dairy products. In this study, *Lactobacillus* strains were isolated from cattle-yak milk for potential probiotic candidates using a series of in vitro tests, including probiotic characterization and safety evaluation (antibiotic susceptibility and hemolytic ability). The results found that the *Lactobacillus*
*rhamnosus* CY12 strain showed a high survival rate in bile salts, under acid conditions, and in the gastrointestinal juice environment, as well as showing high antimicrobial activity and adhesive potential. The safety evaluation showed that all strains were considered non-hemolytic. In addition, the whole-genome sequencing indicated that the strain CY12 spanned 2,506,167 bp, with an average length of 881 bp; the GC content in the gene region (%) was 47.35, contained 1347 protein-coding sequences, and accounted for 85.72% of the genome. The genome annotation showed that genes mainly focused on the immune system process, metabolic process, carbohydrate utilization, carbon metabolism, galactose metabolism, and biological adhesion, etc. This study revealed that the *Lactobacillus*
*rhamnosus* CY12 strain might be an excellent potential probiotic in the development of feed additives for animals and has the ability to promote health.

## 1. Introduction

In the past decades, antibiotic misuse and overuse in humans and livestock has led to the accelerated emergence of antibiotic-resistant organisms and the evolution of antibiotic-resistant strains, which have been serious risks to human security and animal health, as well as to the environment [1]. Consequently, there is a common tendency to reduce or avoid using antibiotics in animal husbandry. Hence, it is crucial to find an ideal substitute for antibiotics. Nowadays, the use of prebiotics as a second-generation substitute for antibiotics has been the hotspot in the research on animal nutrition. Probiotics are “living microorganisms with low or no pathogenicity, which exerts beneficial effects on the host health when ingested in sufficient amounts [2,3]”. As one of the most important bacterial groups in the food industry, the application of lactic acid bacteria (*LAB*) is considered to have beneficial effects on the health of animals and humans [4], such as improved digestion [5], strengthened mucosal immune barrier [6], treatment of diarrhea [7], anti-diabetic effects [8], improved immunity [9], maintenance of intestinal balance [10], management of inflammatory bowel diseases [10], and relieved constipation [11]. However, the mode of action of probiotics on the host is still not fully understood. Therefore, studying the in vitro characteristics and evaluating their probiotic properties are essential for the optimization and use of probiotics.

*LAB* possess inhibitory characteristics against enteropathogens and regulate the intestinal microbial balance, thanks to the production of lactic acid and metabolites, such as antimicrobial compounds, antioxidants, and organic acids [12]. These are considered important criteria in selecting potential probiotic strains because they are considered key ecological factors in determining the dominant bacteria in the intestinal ecosystem [13]. However, in recent years, a growing body of research has revealed that functions of probiotics are highly strain-specific and their biological effects should be individually evaluated [9]. Therefore, it is necessary to isolate and screen novel probiotics strain with potential health benefits. Dairy production systems are still important sources of probiotic strains [14]. The cattle-yak has obvious heterosis, which is a hybrid offspring of yak (*Bos grunniens*) and in cattle (*Bos taurus*) [15]. Cattle-yak milk, as a plateau specialty milk, has high quality, rich nutrition, high vitamin contents, and a fragrant smell. It is not only an important food and a dairy-processing raw material for plateau herdsmen, but is also a high-quality source of nutrients, such as fat, protein, lactose, and inorganic salt. It is expected to isolate probiotic strains with natural and excellent characteristics. However, there are few studies on probiotics. For the selection of a potential beneficial strain, candidate probiotic strains should be characterized by safety, function, and prebiotics, including antimicrobial susceptibility, adhesion to the intestinal mucosa and gastrointestinal transit, tolerance to the GI environment, acid potential, and antagonism against pathogens [13]. In vitro methods are widely used to screen novel beneficial strains because examining in vivo efficacy is expensive and time-consuming. Moreover, it is necessary to analyze and evaluate the related characteristics of their probiotic function at the gene level to explore more potential biological functions and information [16].

Therefore, the aim of this study was to identify the beneficial strains from the milk of cattle-yak, and to evaluate the probiotic characteristics of potential strains through a series of in vitro tests. Additionally, the whole genome sequence was also analyzed to provide insight into probiotic-associated genes for a better understanding of their potential biological functions and information.

## 2. Materials and Methods

### 2.1. Bacterial Strains and Culture Conditions

The *Lactobacillus* (*LAB*) strains used in this study were isolated from the milk of healthy cattle-yak. *LAB* were isolated on de Man, Rogosa, and Sharpe agar (MRS; Guangdong Huankai Microbial Sci. & Tech. Co. Ltd., Guangzhou, China) containing 1.0% CaCO_3_ (wt/vol) and 0.005% crystal violet. The plates were cultured in an anaerobic incubator at 37 °C for 48 h. Isolated colonies that generated surrounding clear zones were selected, and the typical single colony was repeatedly inoculated on MRS agar plates until the colony morphology was consistent. Then, bacteria were identified using Gram staining (Gram-positive) and catalase tests (catalase-negative colonies). The genomic DNA of selected strains was extracted using a DNA extraction kit (Omega Bio-tek, Norcross, GA, USA) according to the manufacturer’s instructions. The quality of the purified DNA samples was checked using an Agilent 2100 Bioanalyzer (Agilent Technologies, Santa Clara, CA, USA), and further PCR-amplified using universal primers for bacteria [17]. The PCR products were identified by 16S rRNA gene sequencing (Tsingke Biotech, Xi’an, Shaanxi, China), and by consulting the NCBI BLAST sequence database (https://blast.ncbi.nlm.nih.gov/Blast.cgi (accessed on 25 October 2021)). Each isolated *Lactobacillus* strain was stored at −80 °C in cryovials containing de Man, Rogosa, and Sharpe broth (MRS; Guangdong Huankai Microbial Sci. & Tech. Co., Ltd., Guangzhou, China) and glycerol (20%). Before performing the experiments, bacteria were subcultured three times, every 16–24 h at 37 °C in MRS broth.

### 2.2. Functional Characteristics of LAB

#### 2.2.1. Antagonistic Activity of the *LAB* Strains

The antagonistic activities were measured by the Oxford cup method as described previously [12]. Pathogenic bacteria were used as follows: *Escherichia coli* (*E. coli*) ATCC 25922, *Staphylococcus aureus* (*S. aureus*) ATCC 29213, *Salmonella typhimurium* (*S. typhii*) ATCC 14028, and *Streptococcus agalactiae* (*S. agalactis*) ATCC 13523. Moreover, *Lactobacillus rhamnosus* (*L. rhamnosus*) ATCC 7469 was used as a contrast. Briefly, the *Lactobacillus* strains were cultured overnight in MRS broth at 37 °C in a shaker. Pathogenic bacteria were cultured in LB (Luria Bertani) broth at 37 °C in a shaker for 24 h. The *LAB* strains were centrifuged at 6000× *g* for 10 min and the supernatant was collected and filtered using a 0.22 µm-pore-size filter (Millipore). Approximately 10^7^ CFU/mL of each pathogen strain was spread onto the surface of LB agar plates. Wells of diameter 5 mm were punctured into the inoculated pates, and 100 µL of supernatant were added into each well and cultured at 37 °C for 24 h. Then, the inhibition zone diameters were measured using a vernier caliper. The experiments were performed in triplicate. The strains with antibacterial activity were used for the subsequent assay.

#### 2.2.2. Evaluation of Growth and Acid-Producing Ability

Growth and acid-producing ability were measured according to a method from a previous study [8]. The strains were inoculated (1%, *v*/*v*) in an MRS broth culture medium and incubated anaerobically at 37 °C for 24 h and shaken at 170 rpm. Viable bacteria were measured with an enzyme-labeling measuring instrument at 600 nm (OD_600_ nm) at approximately 3 h intervals. Meanwhile, the pH of the cultures was measured using a digital pH meter. Three independent experiments were performed for each strain, and the growth curve and acid-producing curve were plotted using MRS broth as a control.

#### 2.2.3. Acidic Resistance and Bile Salts of Isolated Strains

According to the basic method reported by Li et al., acid-tolerance test of isolated strains was carried out [12]. Cells of isolated strains were cultured in MRS broth overnight at 37 °C and propagated twice before the assay. The acid pH of the MRS broths was adjusted to 1.5, 2.5, 3.5, 4.5, and 6.8 using 1 M hydrochloric acid. The activated strains were inoculated under different pH conditions and cultured at 37 °C for 4 h. The bactreial resistance to bile salts was detected following the method described by Nami et al. [18]. The cultured strains were inoculated in MRS broths containing 0, 0.1%, 0.2%, 0.3%, and 0.5% bile salts incubation for 4 h at 37 °C. Moreover, the OD_600_ was evaluated using a microplate reader and the survival rate was calculated based on pH 6.8 as a contrast using the formula below:$$\mathrm{Survival}\;\mathrm{rate}\left(\%\right)={\mathrm{OD}}_{600}\left(\mathrm{experiment}\right)/{\mathrm{OD}}_{600}\left(\mathrm{control}\right)\times 100$$

#### 2.2.4. Tolerance to Gastrointestinal (GI) Conditions

The in vitro model established as described by Li et al. was used to simulate gastric juice and intestinal juice [9], with slight modifications. The artificial gastric juice was prepared by adding 3 g/L pepsin (Solarbio, Beijing, China) into sterile 20 mL 1 mol/L HCL, sterilized with 0.22 µm filter membrane and adjusted pH to 2.0, 2.5, and 3.0 with sterile 1 mol/L NaOH. The artificial intestinal juice was prepared by adding 1.0 g/L trypsin (Solarbio, Beijing, China) and 0.27 g KH_2_PO_4_ into sterile 20 mL dH_2_O, pH was adjusted to 6.8 by 1 mol/L NaOH, and filtered by 0.22 µm membrane. The *Lactobacillus* strains were activated overnight in MRS broth at 37 °C in the shaker. Bacterial solution was centrifuged at 6000× *g* for 10 min and the strains were obtained, washed with sterile PBS three times, and resuspended in 4% sterile normal saline to prepare the bacterium suspension. The 1 mL bacterium suspension was inoculated in 5 mL simulated gastric juice at pH 2.0, 2.5, and 3.0, and incubated for 0, 1, 2, and 3 h at 37 °C. The 1 mL bacterial suspension was inoculated into 5 mL simulated intestinal fluid for 0, 2, 4, 6, and 8 h, and then cultured on MRS agar for 24 h to determine the tolerance of selected strains to artificial gastrointestinal fluid, using the formula below:$$\mathrm{Survival}\;\mathrm{rate}\left(\%\right)=\log\mathrm{CFU}\left(N_{t}\right)/\log\mathrm{CFU}\left(N_{0}\right)\times 100\%$$
where $N_{0}$ is the total viable counts of the selected strains at 0 h, and $N_{t}$ is the total viable counts after exposure to the artificial gastrointestinal juice during the different time periods.

#### 2.2.5. Antibiotic Susceptibility of Selected Strains

According to the guidelines of the Institute of Clinical and Laboratory Standards Institute (CLSI), the disk diffusion method was used to test the susceptibility of selected strains [19]. Susceptibility to the following thirteen antibiotics was assessed: 10 µg of ampicillin, penicillin, gentamicin, and amoxicillin; 30 µg each of tetracycline, chloramphenicol, kanamycin, cefazolin, and vancomycin; 15 µg of erythrocin; 1 µg of oxacillin; and 5 µg of ciprofloxacin. A volume of 200 µL of each strain (1 × 10^6^ CFU/mL) was spread on MRS agar plates, and paper discs were set on the plate, and cultured at 37 °C for 24 h. After the culturing, inhibition zone diameters (cm) were measured using a vernier caliper. The criteria are described as follow: S = sensitive (zone diameter ≥ 1.7 cm); I = intermediate (zone diameter 1.2 to 1.7 cm); R = resistant (zone diameter ≤ 1.2 cm) [20]. The assays were performed in triplicate with 3 independent experiments.

#### 2.2.6. Hemolytic Activity Analysis

The selected strains were inoculated in MRS broth overnight at 37 °C and shaken at 170 rpm. The *LAB* strains were spread on Columbia blood plate medium (Dijing Microbial Technology Co., LTD, Guangzhou, China) using the streak plate method, and then the plates were incubated at 37 °C for 48 h. The appearance of a transparent or semitransparent zone around the bacterial colonies was indicated as positive hemolytic activity.

#### 2.2.7. Adhesion to Human Colon Carcinoma (Caco-2) Cells

The adhesion of selected strains to Caco-2 cells was estimate by the previous method, but modified slightly [20]. Caco-2 cells were cultured in Minimum Essential Medium (MEM) and supplemented with 10% (*v*/*v*) heat-inactivated fetal bovine serum (FBS), 1% MEM Non-Essential Amino Acids (NEAA), 1% Sodium Pyruvate (SP), and 1% L-glutamine (all reagents were manufactured by Gibco, and obtained from Thermo Fisher Scientific, Shanghai, China). The Caco-2 cells were incubated in an incubator containing 5% CO_2_ and 95% air at 37 °C for 3 to 5 d. The cells were washed when they reached 80 to 90% confluence with sterile PBS, and split using trypsin (Gibco, Thermo Fisher Scientific, Shanghai, China). The Caco-2 cells were plated into 12-well plates at a concentration of 10^5^ cells/well and cultured at 37 °C for 24 h. The selected strains were cultured in MRS broth (3% *v*/*v*) at 37 °C overnight before the experiment. The selected strains were centrifuged at 6000× *g* for 10 min and washed with sterile PBS three times, and resuspended in non-supplemented MEM at a concentration of 10^8^ CFU/mL. The Caco-2 cells were washed using sterile PBS, and 1 mL bacterial broth (10^8^ CFU/mL) was added to each well and co-cultured at 37 °C for 3 h. The supernatant was aspirated to remove non-adhering bacteria. The cells were washed five times using sterile PBS. Moreover, each well was added with 1 mL of 1% (*v*/*v*) Triton X-100 in sterile PBS and was cultured at 4 °C for 30 min. The cells were obtained and transferred to sterilized 1.5 mL tubes, centrifuged at 6000× *g* for 10 min, and washed twice with sterile PBS. Finally, the cells were resuspended in sterile PBS and continuous aseptic dilution, which were used to inoculate the MRS agar plates, and were cultured for 48 h at 37 °C to assess the adherence ability. The calculation formula of adhesion rate was below:$$\mathrm{Adhesion}\;\mathrm{rate}\left(\%\right)=\log\mathrm{CFU}\left(N_{t}\right)/\log\left(N_{0}\right)\times 100\%$$
where $N_{0}$ is the total viable counts of *LAB* strains before treatment and $N_{t}$ is the total viable counts of adherent bacteria.

### 2.3. Whole-Genome Sequencing

#### 2.3.1. DNA Extract and Genome Sequence

Following the manufacturer’s instructions, the genomic DNA of strain CY12 was extracted from the cell pellets using Bacteria DNA Kits (Omega Bio-tek, Norcross, GA, USA). The quality of the purified DNA samples was checked using an Agilent 2100 Bioanalyzer (Agilent Technologies, Santa Clara, CA, USA). The quantification of genomic DNA was carried out using a TBS-380 fluorometer (Turner Biosystems Inc., Sunnyvale, CA, USA). A high-qualified DNA sample (OD_260/280_ = 1.8~2.0, >6 µg) was utilized to construct a fragment library. The whole genome of strain CY12 was sequenced using the Pacific Biosciences Sequel platform and the Illumina Hiseq platform (PE150 mode). At the suggestion of the manufacturer, the sequencing libraries were purified three times using 0.45 × volumes of Agencourt AMPure XPbeads (Beckman Coulter Genomics, Danvers, MA, USA).

#### 2.3.2. Genome Assembly

Raw sequencing data were generated by Illumina base-calling software CASAVA v1.8.2 (http://support.illumina.com/sequencing/sequencing_software/casava.ilmn (accessed on 18 September 2021)) according to its corresponding manuscript. Trimmomatic (http://www.usadellab.org/cms/uploads/supplementary/Trimmomatic (accessed on 18 September 2021)) uses default parameters to identify contamination reads, such as those containing adaptors or primers. Clean data obtained by the above quality control processes were further analyzed. The CY12 genome was sequenced using a combination of the PacBio RS and Illumina sequencing platforms. The Illumina data were used to assess genomic complexity and correct the PacBio long reads.

#### 2.3.3. Genome Annotation

We used the ab initio prediction method for the prokaryotic organism to obtain gene models for strain CY12. Gene models were identified using Glimmer 3 [21]. Then, all gene models were blasted against the non-redundant (NR in NCBI) database, SwissProt (http://uniprot.org (accessed on 18 September 2021)), KEGG (http://www.genome.jp/kegg/ (accessed on 18 September 2021)) [22], and COG (http://www.ncbi.nlm.nih.gov/COG (accessed on 18 September 2021)) [23] to conduct functional annotation by the blastp module. In addition, tRNA was identified using the tRNAscan-SE (v1.23, http://lowelab.ucsc.edu/tRNAscan-SE (accessed on 18 September 2021)) [24] and rRNA [25] was determined using the RNAmmer (v1.2, http://www.cbs.dtu.dk/services/RNAmmer/ (accessed on 18 September 2021)). Meanwhile, Circos v0.64 (http://circos.ca/ (accessed on 18 September 2021)) was used to map the genome of the strain CY12.The phylogenetic tree of strain CY12 was constructed using MEGA5 software.

### 2.4. Statistical Analysis

SPSS 19.0 (SPSS Inc., Chicago, IL, USA) was used to carry out one-way analysis of variance (ANOVA) and to compare the significant differences among different treatments. When the data did not follow a normal distribution, Duncan’s multiple ranges were performed, and *p* < 0.05 was considered to be statistically significant.

## 3. Results

### 3.1. Molecular Identification of Lactobacilli Strains

In the present study, forty-five strains were isolated from the milk of cattle-yak according to the clearing zone around the colony on MRS agar plates, to which were added 1% CaCO_3_, catalase-negative test, and Gram-positive staining (Appendix A). Among these strains, only seven strains had antagonistic activity at varying degrees. Therefore, the seven selected strains were further confirmed using the 16S rRNA technique. The results of the 16S rRNA sequencing demonstrated that the seven strains were *Lactobacillus plantarum* (CY2; 99.24%), *Lactobacillus paracasei* (CY3; 97.77%), *Streptococcus thermophilus* (CY7; 100.00%), *Streptococcus thermophilus* (CY11; 99.71%), *L. rhamnosus* (CY12; 99.69%), *Enterococcus faecalis* (CY12-2; 95.89%), and *Enterococcus faecalis* (CY13; 92.31%). The 16S sequences were deposited in the National Center for Biotechnology Information (NCBI) with the accession numbers OK605898–OK605904.

### 3.2. Functional Characteristics of LAB

#### 3.2.1. Antagonistic Activity

Out of the forty-five *Lactobacillus* strains, only seven strains had antagonistic activity at varying degrees (Table 1). The results showed that CY12, CY2, and ATCC 7469 had higher antagonistic activity against pathogenic bacteria, including *E. coli*, *S.aureus*, *S.typhii,* and *S.agalactis,* than against other stains (*p* < 0.001). Other *LAB* strains showed weaker antagonistic effects. Consequently, seven strains were selected for further experimentation.

#### 3.2.2. Growth and Acid-Producing Ability

Growth curves of the seven strains in MRS broth were constructed using sterile MRS medium and ATCC 7469 as controls (Figure 1A). The concentrations of the seven strains gradually increased at varying degrees with the growth time. The *LAB* strains of CY2, CY3, CY12, CY12-2, and ATCC 7469 entered the exponential growth stage about 3 h after inoculation, while the strains of CY7, CY11, and CY13 showed the characteristic of slow growth. The stationary phase of all *LAB* strains reached approximately 15 h after inoculation. After reaching the stationary phase, the growing concentration of *LAB* CY12 was the highest, followed by ATCC 7469, with a final OD_600nm_ of 1.375 and 1.157, respectively, whereas CY7, CY11, and CY13 reached an OD_600nm_ of 0.510, 0.430, and 0.442, respectively.

Figure 1B shows the acid-producing ability of the studied strains. The acid-producing capacity of the seven strains gradually increased at varying degrees with the fermentation time, and the stationary phase of all *LAB* strains was reached approximately 18 h after culture. The acid-producing capacity of the CY12 strain was higher than for the other strains, with a final pH of 3.78, while CY7 had the lowest acid-producing capacity and pH was 4.80.

#### 3.2.3. Acid and Bile Salt Tolerance of *LAB* Isolates

The acidic and bile salt conditions had different effects on the survivability of all *LAB* strains (*p* < 0.05). Table 2 and Table 3 show the survival rate of the tested strains at different stages. The survival percentage of all strains decreased with the decrease of pH value and the increase of bile salt concentration. All strains were able to survive at pH 1.5, but no significant difference was observed for each strain at pH 1.5 (*p >* 0.05). However, the *LAB* strains exhibited tolerance at pH 2.5, 3.5, and 4.5 (*p* < 0.05). The CY2, CY12, and ATCC 7469 strains were observed to have higher survival rates than other *LAB* strains. Moreover, the CY2 and CY12 strains had higher survivability compared to the control and other stains at pH 2.5 (*p* < 0.001).

The effects of bile salt at different concentrations on studied strains are listed in Table 3. The results showed that the resistance ability of *Lactobacillus* strains gradually decreased with the increase in concentration. All strains were resistant to 0.1% bile salt with an over 40% survival rate, except CY7 and CY12-2. The CY12 strains were exposed to bile salt at 0.3% for 4 h; their survival percentage was superior to the control strain ATCC 7469 and other strains (*p* < 0.001), while all strains were inferior to that of the control strain ATCC 7469 at 0.2%. Furthermore, at the 0.5% bile salt concentration, the selected strains had lower survivability, with a less than 20% survival rate, except CY2, CY12, and ATCC 7469 strains.

#### 3.2.4. Survival under Simulated Gastrointestinal Conditions

The survivability in the gastrointestinal tract is an important characteristic needed for probiotics. We assessed the simulated gastric juice tolerance of seven *LAB* strains at pH 2.0, 2.5, and 3.0, among which seven strains presented good tolerance (Table 4). The result showed that the survival percentage of the selected strains gradually decreased with increasing time at different concentrations. Notably, in the simulated gastric environment at pH 2.0, only CY2, CY12, and ATCC 7469 strains showed high tolerance (survival percentage >80%) at pH 2.0 for 2 h. Moreover, the survival rate of the CY12 strain was superior to the control ATCC 7469 at pH 2.0 for 2 h (survival rate 80.08%, *p* < 0.001). All strains were observed to have better tolerance abilities to artificial gastric juice at pH 2.5 and pH 3.0 in different times. The survival rate of the seven strains ranged from 51.44 to 98.93% in artificial gastric juice.

The effect of artificial intestinal juice on *LAB* strains is shown in Table 5. We found that all of the tested strains were resistant to the simulated intestinal environment, with over 40% survival. The strains CY2 and CY12 were exposed to simulated intestinal juice for 2, 4, 6, and 8 h, and their survival rates were higher than for the control strain ATCC 7469 (*p* < 0.001); the survival rate was >90%. The survival rate of the other *LAB* strains ranged from 41.17 to 56.87%.

#### 3.2.5. Antibiotic Susceptibility

To determine the safety of *LAB* strains, the antibiotic susceptibility of seven strains was assessed according to the Clinical and Laboratory Standards Institute (CLSI) guidelines, and the results are listed in Table 6. All studied strains were susceptible to antibiotics such as penicillin, chloramphenicol, erythrocin, tetracycline, ciprofloxacin, amoxicillin, ampicillin, and cefazolin. However, CY2 and CY13 strains were intermediates of tetracycline and ciprofloxacin, the strains of CY12-2 and ATCC 7469 were intermediates of cefazolin, and CY7 strain was resistant to amoxicillin and ampicillin. Moreover, our results showed that all selected strains were resistant to oxacillin and vancomycin. The *LAB* strains of CY11, CY12, and CY12-1 were intermediate to kanamycin, and other strains were resistant to kanamycin. In addition, the strains of CY2, CY11, and CY13 were resistant to gentamicin. However, other studied strains were intermediate to gentamicin.

#### 3.2.6. Hemolytic Activity

The *LAB* strains showed no hemolytic effect on the blood agar plates (Appendix A). All strains were considered non-hemolytic.

#### 3.2.7. Adherence Ability

The adhesion to Caco-2 cells serves as an important indicator in selecting probiotic strains, with a crucial role in probiotic characteristics. In this present study, the adhesion capacities of selected strains were evaluated and *L. rhamnosus* GG ATCC 7469 was used as a control strain. The results are presented in Figure 2. The four strains exhibited good adhesion ability to Caco-2 cells, and showed more than 60% adhesion to Caco-2 intestinal cells. Meanwhile, the strain CY12 exhibited a higher adhesion capacity than the other strains (*p* < 0.001), and the adhesion percentage was 86.7%, while the strains CY7 and CY11 showed lower adhesion rates i.e., 13.5% and 20.1%, respectively.

Overall, compared to other selected strains, the CY12 strain showed a high survival rate in bile salts, under acid conditions, and in the gastrointestinal juice environment, as well as showing high antimicrobial activity and adhesive potential. Hence, the CY12 strain was selected as the next experimental research object to further explore the potential biological functions and information.

### 3.3. Whole-Genome Sequencing

The whole-genome sequencing was carried out to understand probiotic characteristics and explore the potentials of the strain CY12, the complete circular genome map of strain CY12 is shown in Figure 3A. Based on 16S rRNA sequencing, CY12 showed high sequence identity (≥99%) to the *L. rhamnosus* strain deposited in NCBI. A phylogenetic tree was constructed with 16S rDNA sequences using the neighbor-joining method in MEGA5 software (Appendix A). Furthermore, the general genomic information, including size, gene number, gene length, CDS, and GC %, of the strain CY12 is shown in Table 7.

A total of 1699 GO (Gene Ontology) functions of the genes of CY12 were annotated (Figure 3B), which are mainly focused on biological processes, cellular components, and molecular function. The GO function was annotated on nitrogen utilization, immune system process, biological regulation, metabolic process, response to stimulus, carbohydrate utilization, cellular process, and biological adhesion. Moreover, cell part, cell junction, binding, transporter activity, transcription regulator activity, and molecular function regulator were also significantly enriched. The KEGG (Kyoto Encyclopedia of Genes and Genomes) analysis showed 20 remarkably enriched pathways for genes, represented in Figure 3C. A total of 1449 genes were enriched in KEGG pathways, and the highest levels of enrichment were in metabolic pathways, biosynthesis of secondary metabolites, microbial metabolism in diverse environments, phosphotransferase system (PTS), biosynthesis of amino acids, and carbon metabolism. Meanwhile, galactose metabolism, pyrimidine metabolism, pentose phosphate pathway, and glycine, serine and threonine metabolism were also enriched. In addition, the COG (Clusters of Orthologous Groups of proteins) functional classification of strain CY12 genome was predicted, and the results are shown in Figure 3D. The results of COG classification showed that strain CY12 was mainly involved in the following aspects: (C) energy production and conversion, (E) amino acid transport and metabolism, (G) carbohydrate transport and metabolism, (H) coenzyme transport and metabolism, (M) cell wall/membrane/envelope biogenesis, and (M) secondary metabolites biosynthesis, transport, and catabolism.

## 4. Discussion

Improving human and animal health with probiotics has been the subject of intense scholarly debate. Numerous studies have shown that probiotics may adhere and survive in the gastrointestinal tract of humans and animals, and contribute to maintaining a microecological balance, promoting digestive and metabolic processes, as well as modulating the immune response, thereby enhancing host immunity, and improving human and animal health [9,13,26]. However, because the effectiveness of probiotics is species or strain dependent, they should meet a series of characteristics, such as safety (antimicrobial susceptibility and hemolytic activity), functional (acid and bile salt tolerance and intestinal mucosa adhesion), as well as beneficial (antagonistic activity, growth, and acid-producing ability) characteristics [12,13,17]. Hence, in this study, we focused on beneficial strains with potential probiotic properties isolated from cattle-yak milk using a series of in vitro tests, and the whole-genome sequencing was performed to reveal their potential biological functions and information.

Antimicrobial activity is one of the important criteria for selecting novel potential probiotic strains [26]. Previous studies have shown that *LAB* possesses extremely inhibitory characteristics against foodborne pathogenic bacteria [20,27,28]. Recent studies also suggest that *LAB* strains from milk had different degrees of inhibitory ability against both Gram-negative and Gram-positive pathogens [29,30], in agreement with our results. In this study, the strains C4 and C5 showed no zones of inhibition, while *LAB* strains CY2 and CY12 showed better antimicrobial activity against *E. coli* (29 mm; 28 mm), *S. aureus* (24 mm; 27 mm), *S.agalactis* (27 mm; 29 mm), and *S. typhii* (23 mm; 25 mm) than ATCC 7469 and the other strains. The potential antimicrobial mechanisms may include forming the immune system, competing nutrients, as well as producing organic acids, bacteriocins, and antioxidants, etc. Moreover, antimicrobial activity related to the growth rate and acid-producing ability of *Lactobacillus* are very crucial indicators to select potential beneficial microorganisms, which must have high acid-producing ability and growth velocity to compete for adhesion sites and nutrients within the GI ecosystem, further exerting their probiotic effects and inhibiting pathogenic bacteria [31,32]. Results of the present study suggest that the strain CY12 had a growth rate and acid-producing ability superior to that of ATCC 7469 and the other strains, consistent with our antimicrobial activity assay.

The presence of bile salts and high acidic conditions constitute the biggest barriers to the survival of *Lactobacillus* in the gastrointestinal tract of the host [12,33]. Consequently, the capacity to tolerate both bile salts and acid conditions are two crucial characteristics considered for a candidate beneficial strain [18,34]. A previous study isolated *L. rhamnosus* Z5 from human breast milk and reported that the strain had super resistance to bile salts and acid, but its growth was prominently inhibited at higher concentrations of bile salts [35]. Another study also reported that the growth of *L. rhamnosus GG* was significantly inhibited by bile salts at 0.15% [36]. These findings were in agreement with our results: in the present study, we observed that the strains CY2, CY12, and ATCC 7469 had higher survivability at higher concentrations of bile salts and low pH, but were significantly inhibited at higher concentrations of bile salts, indicating that the strains CY2 and CY12 could survive in the GI tract. High osmotic pressure and surfactant activity of bile salts are potential mechanisms for disrupting phospholipids and proteins in the bacterial cell membrane, causing disruption of the bacterial membrane and causing DNA damage or death, further reducing the strains’ survival [37,38]. However, it has been reported that when *L. rhamnosus* JL-1 was exposed to bile salts for 4 h, the strain had a survival rate as high as 90.01% [9]; this may be due to the *LAB* being isolated from the intestinal tract, and could be more adaptable to the GI ecosystem than other *Lactobacillus* [12].

A crucial property of *Lactobacillus* intended for the oral route of administration is the capability to survive the gastric and intestinal conditions, including the presence of lysozyme, acid pH values with pepsin, pancreatin, and bile salts [39,40]. From this study, The results showed that all the *Lactobacillus* strains had good tolerance to pepsin and trypsin simulated gastrointestinal juice, and the strains CY2 and CY12 showed high tolerance (survival percentage >80%) to artificial gastric juice at pH 2.0 for 2 h. Meanwhile, they also had better tolerance abilities to simulated intestinal juice for 2, 4, 6, and 8 h. In contrast, the survival percentages of the other strains decreased significantly. The survival rates of CY2 and CY12 in the gastrointestinal juice conditions were higher than the survival percentage of previously reported LAB, such as *L. rhamnosus* and *Lactobacillus casei* [35,41,42]. Another study reported that the *LAB* showed strong tolerance to the simulated GI environment, with survival rates of >80% at pH 2.0 to 3.0, which is consistent with our results [43].

It has been reported that candidate probiotics should not serve as antibiotic resistance genes for hosts [31,44]. Therefore, the assessment of antimicrobial sensitivity is considered essential for the safe use of potential probiotics. From this study, we evaluated the antimicrobial susceptibility of seven selected strains using the disk diffusion method, and found that all studied strains were susceptible to antibiotics such as penicillin, chloramphenicol, erythrocin, tetracycline, ciprofloxacin, amoxicillin, ampicillin, and cefazolin, supporting most of the *LAB* strains exhibited sensitivity to tetracycline, penicillin, and chloramphenicol, indicating that lactic acid bacteria generally had low resistance to these antibiotics [13,31,45]. However, we observed that all selected strains are resistant to oxacillin and vancomycin. It is well known that *Lactobacillus* strains are usually resistance to β-lactam antibiotics such as oxacillin, ceftriaxone, and ampicillin, due to the presence of β-lactamase in lactic acid bacteria [26,28]. Likewise, resistance to glycopeptides (vancomycin) has been noted in *LAB* and is associated with congenital resistance to membrane permeability (in most cases), possibly through a mechanism of resistant efflux [29]. Similar results found from previous studies also showed resistance to kanamycin, vancomycin, and ampicillin [44,46,47]. In addition, hemolytic activity is considered to be an important indicator of the safety of potential beneficial strains; microbial probiotic strains are not expected to dissolve red blood cells when ingested in humans or animals. Like most *LAB* strains [48], our results found that all strains showed non-hemolytic activity and, thus, are safe for use. However, it is necessary to further perform in vivo safety assays.

The adhesion to the mucosal surfaces and epithelial cells is considered a crucial characteristic for candidate probiotics selection and plays an important role in competitively excluding or inhibiting pathogens [26,49]. Most *Lactobacillus* produce cell surface proteins and help bacteria bind to the gastrointestinal epithelium, thereby improving immunity and antagonism against pathogens [50]. The previous study demonstrated that the *L. rhamnosus* GG strain had a higher adherence capacity to Caco-2 cells [13]; similar results have been observed in the present study, whereby the strain CY12 exhibited higher adhesion capacity than the other strains, with over 80% adhesion capacity. However, whether the selected strains could inhibit invasion and intracellular survival of bacteria needs to be further researched. In this study, the CY12 strain was selected according to its probiotic features and safety evaluation. Genome-wide sequencing was performed on the *L. rhamnosus* CY12 strain to better understand its potential biological functions. According to the definition of bacterial genome size by Ochman and Davalos [51], the genome size of *L. rhamnosus* CY12 isolated in this study is 2,506,167 bp, which indicates a medium-sized genome; this type of bacteria usually has a strong metabolic ability and tolerance [52] and can adapt to different ecological environments. Moreover, based on the GO, KEGG, and COG annotation results, we also found that the genes were outstanding in the immune system process, biological adhesion, metabolic pathways, and galactose metabolism; glycine, serine, and threonine metabolism; energy production and conversion; amino acid transport and metabolism; and carbohydrate transport and metabolism, indicating that the strains were involved in the immune system process, adhesion, and had a metabolic ability and tolerance [16]. These results are consistent with our noted beneficial characteristics of the *L. rhamnosus* CY12 strain. However, there were still some genes for which we were unable to obtain COG functional annotation, which requires us to further explore these unknown functional genes in future studies.

## 5. Conclusions

In conclusion, among the seven *LAB* strains, *L. rhamnosus* CY12 showed the best properties to be used as a potential probiotic candidate, as demonstrated by the antimicrobial activity, the growth and acid-producing ability, antibiotic susceptibility, adherence ability, hemolytic ability, and survivability in bile salts, under acid conditions, as well as in the gastrointestinal tract. Moreover, whole-genome information of *L. rhamnosus* CY12 was obtained by gene assembly. The strain CY12 spanned 2,506,167 bp with an average length of 881 bp; the GC content in the gene region (%) was 47.35, containing 1347 protein-coding sequences, and accounted for 85.72% of the genome. The genome annotation showed that genes mainly focused on the immune system process, biological adhesion, metabolic pathways, and galactose metabolism; glycine, serine and threonine metabolism; energy production and conversion; amino acid transport and metabolism; and carbohydrate transport and metabolism. These findings support the potential application of *L. rhamnosus* CY12 in human or animal food processing and drug production in the future as a potential probiotic candidate. Nevertheless, further in vivo tests are needed to evaluate their potential health benefits before their utilization as probiotics in functional foods.

## Figures and Tables

**Figure 1 biology-11-00044-f001:**
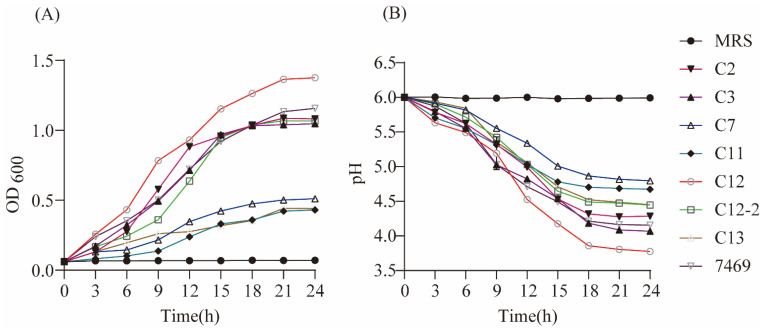
Growth and acid-producing ability of selected strains. (**A**) Growth curve of selected strains; (**B**) acid-producing curve of selected strains.

**Figure 2 biology-11-00044-f002:**
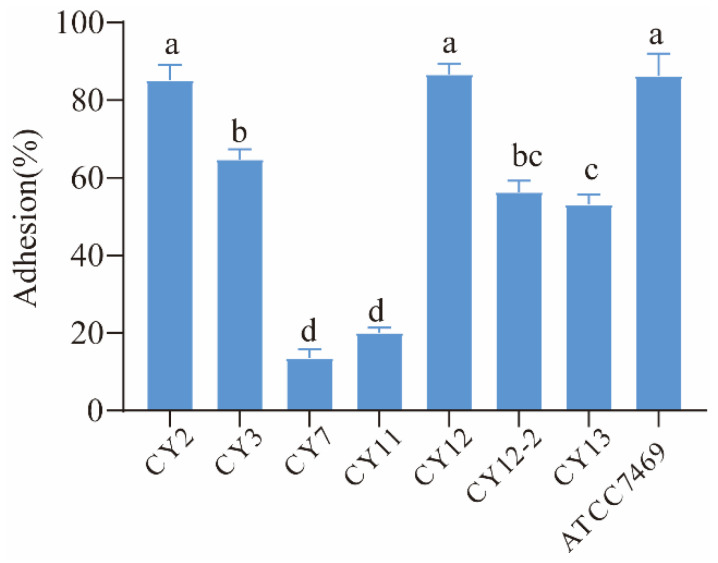
Adhesion rate of selected strains. a–d means different superscript letters are different per *p* < 0.05.

**Figure 3 biology-11-00044-f003:**
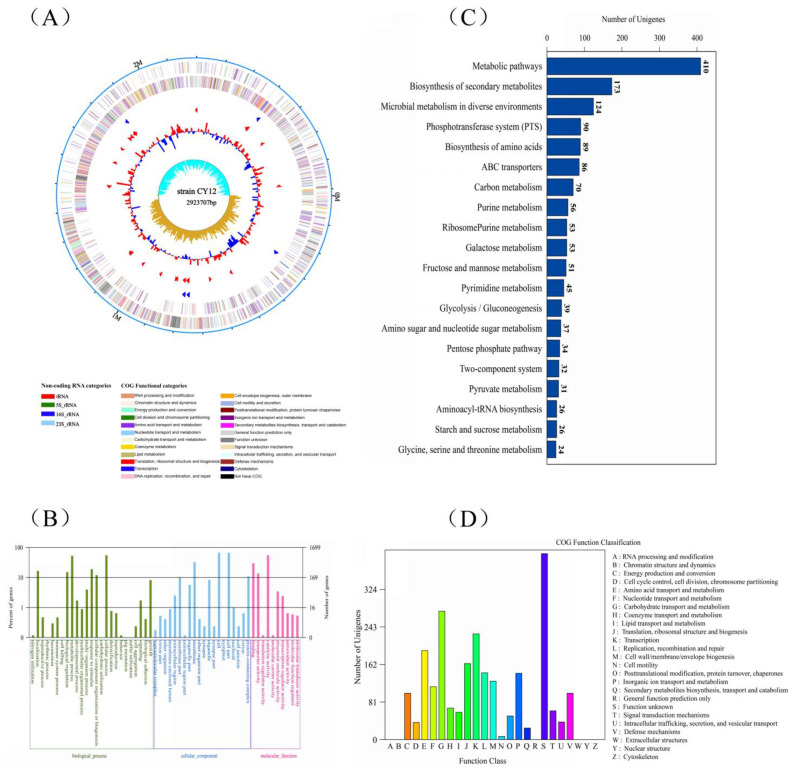
Whole-genome sequencing of selected strains. (**A**) The complete circular genome map of strain CY12; the outermost circle of the circle diagram is the marker of genome size, and each scale is 0.5 MB; the second and third circles are CDS on the positive and negative chains. Different colors indicate the functional classification of different COG of CDS. The fourth circle is rRNA and tRNA; the fifth circle is GC content. The outward red part indicates that GC content in this region is higher than the average GC content in the whole genome. The higher the peak value, the greater the difference between GC content in this region and the average GC content. The innermost circle is the GC skew value, and the specific algorithm is G-C/G + C. In the biological sense, when the value is positive, the positive chain is more inclined to transcribe CDS; when the value is negative, the negative chain is more inclined to transcribe CDS (the form of the circle graph is flexible, and the above is only the most traditional form); (**B**) gene ontology (GO) analysis of strain CY12 genome; (**C**) KEGG pathways enrichment for strain CY12 genome; (**D**) the Clusters of Orthologous Groups of proteins (COG) functional classification of the CY12 strain genome.

**Table 1 biology-11-00044-t001:** Pathogen inhibition by *Lactobacillus* supernatant.

Indicator Strains	*E. coli*	*S. aureus*	*S. agalactis*	*S. typhii*
CY2	29.19 ^a^	24.59 ^ab^	27.33 ^ab^	23.87 ^b^
CY3	15.13 ^cd^	13.20 ^cd^	15.07 ^cd^	12.03 ^de^
CY7	15.10 ^c^	15.00 ^c^	15.37 ^cd^	12.93 ^d^
CY11	14.13 ^e^	12.89 ^cd^	13.47 ^c^	11.27 ^e^
CY12	28.27 ^ab^	27.40 ^a^	29.23 ^a^	25.13 ^a^
CY12-2	15.233 ^cd^	14.10 ^c^	16.83 ^c^	13.43 ^d^
CY13	14.50 ^de^	11.57 ^cd^	16.23 ^c^	11.23 ^e^
ATCC 7469	27.70 ^ab^	22.63 ^b^	26.77 ^b^	22.87 ^c^
SEM	0.13	0.24	0.15	0.09
*p*-value	<0.001	<0.001	<0.001	<0.001

Data are presented as means ± SEM. *E. coli*, *Escherichia coli*; *S. aureus*, *Staphylococcus aureus*; *S. typhii*, *Salmonella typhimurium*; *S. agalactis*, *Streptococcus agalactiae*. SEM, standard error of the mean; ^a–e^ means within columns with different superscript letters are different per *p* < 0.05.

**Table 2 biology-11-00044-t002:** Effect of pH on viability of *Lactobacillus*.

Indicator	pH 1.5	pH 2.5	pH 3.5	pH 4.5
CY2	13.10	96.97 ^a^	97.62 ^a^	98.84 ^a^
CY3	12.03	36.09 ^c^	43.11 ^b^	44.58 ^b^
CY7	10.91	9.68 ^e^	13.66 ^d^	22.03 ^c^
CY11	10.73	10.90 ^de^	11.62 ^d^	25.64 ^c^
CY12	14.69	98.08 ^a^	98.27 ^a^	98.02 ^a^
CY12-2	10.30	10.02 ^de^	40.61 ^bc^	40.03 ^b^
CY13	13.28	12.79 ^d^	38.53 ^c^	43.79 ^b^
ATCC 7469	13.79	93.01 ^b^	95.16 ^a^	97.77 ^a^
SEM	0.87	0.72	1.04	2.77
*p*-value	0.058	<0.001	<0.001	<0.001

Data are presented as means ± SEM. pH 1.5, pH 2.5, pH 3.5, and pH 4.5 represent pH values 1.5, 2.5, 3.5, and 4.5, respectively. SEM, standard error of the mean; ^a–e^ means within columns with different superscript letters are different per *p* < 0.05.

**Table 3 biology-11-00044-t003:** Effect of bile salt on viability of *Lactobacillus*.

Indicator	0.1%	0.2%	0.3%	0.5%
CY2	58.28 ^a^	22.46 ^c^	21.43 ^b^	20.10 ^a^
CY3	44.51 ^ab^	15.57 ^d^	14.92 ^c^	12.38 ^b^
CY7	20.04 ^c^	16.27 ^d^	15.61 ^c^	14.72 ^b^
CY11	48.89 ^ab^	16.46 ^d^	15.47 ^c^	14.87 ^b^
CY12	56.56 ^a^	35.42 ^b^	28.38 ^a^	22.27 ^a^
CY12-2	27.70 ^c^	15.63 ^d^	14.91 ^c^	13.85 ^b^
CY13	41.33 ^ab^	15.81 ^d^	13.62 ^c^	13.56 ^b^
ATCC 7469	51.47 ^ab^	44.90 ^a^	22.96 ^b^	20.60 ^a^
SEM	1.60	0.86	1.09	1.03
*p*-value	<0.001	<0.001	<0.001	0.003

Data are presented as means ± SEM. 0.1%, 0.2%, 0.3%, and 0.5% represent the concentrations of bile salt at 0.1%, 0.2%, 0.3%, and 0.5%, respectively. SEM, standard error of the mean; ^a–d^ means within columns with different superscript letters are different per *p* < 0.05.

**Table 4 biology-11-00044-t004:** Effect of simulated gastric juice on the viability of *Lactobacillus*.

Time (h)	pH 2.0			pH 2.5			pH 3.0		
	1	2	3	1	2	3	1	2	3
CY2	83.33 ^ab^	80.35 ^a^	41.93 ^d^	95.69 ^a^	91.21 ^a^	89.47 ^a^	98.93 ^a^	97.76 ^a^	95.94 ^a^
CY3	89.61 ^a^	74.93 ^c^	67.83 ^b^	92.27 ^a^	90.74 ^b^	87.08 ^a^	96.51 ^a^	95.68 ^a^	94.78 ^a^
CY7	70.45 ^c^	37.11 ^e^	33.23 ^e^	74.35 ^c^	79.33 ^c^	51.44 ^e^	96.59 ^a^	83.93 ^c^	78.61 ^c^
CY11	86.23 ^ab^	52.01 ^d^	50.48 ^c^	91.19 ^a^	76.53 ^c^	62.83 ^d^	96.69 ^a^	85.97 ^bc^	78.66 ^c^
CY12	83.66 ^ab^	86.09 ^a^	80.08 ^a^	94.59 ^a^	93.56 ^a^	85.85 ^a^	96.62 ^a^	95.55 ^a^	88.90 ^b^
CY12-2	78.61 ^b^	73.59 ^c^	68.13 ^b^	83.33 ^b^	77.70 ^bc^	71.14 ^c^	89.05 ^b^	93.80 ^ab^	88.61 ^b^
CY13	82.67 ^ab^	39.94 ^e^	39.43 ^de^	91.46 ^a^	76.14 ^c^	88.98 ^a^	93.70 ^b^	76.38 ^c^	73.67 ^d^
ATCC 7469	88.06 ^ab^	87.41 ^a^	76.44 ^a^	92.67 ^a^	82.62 ^b^	80.07 ^b^	96.53 ^a^	91.58 ^b^	81.89 ^c^
SEM	2.46	1.02	2.13	1.46	2.37	1.86	1.60	3.24	1.29
*p*-value	<0.001	0.002	0.001	<0.001	<0.001	<0.001	<0.001	<0.001	<0.001

Data are presented as means ± SEM. pH 2.0, pH 2.5, and pH 3 represent pH values 2, 2.5, and 3, respectively. Numbers 1, 2, and 3 represent the incubated times of 1 h, 2 h, and 3 h, respectively. SEM, standard error of the mean; ^a–e^ means within columns with different superscript letters are different per *p* < 0.05.

**Table 5 biology-11-00044-t005:** Effect of simulated intestinal juice on the viability of *Lactobacillus*.

Indicator	Time (h)			
	2	4	6	8
CY2	98.39 ^a^	95.47 ^ab^	91.87 ^ab^	90.87 ^a^
CY3	56.12 ^b^	55.98 ^c^	54.91 ^c^	52.47 ^c^
CY7	56.87 ^b^	56.02 ^c^	55.57 ^c^	55.02 ^c^
CY11	54.92 ^b^	53.66 ^c^	52.20 ^c^	50.02 ^c^
CY12	97.38 ^a^	92.81 ^a^	92.73 ^a^	89.66 ^a^
CY12-2	43.96 ^b^	43.42 ^c^	42.71 ^c^	41.17 ^c^
CY13	54.74 ^b^	54.46 ^c^	52.32 ^c^	51.70 ^c^
ATCC 7469	92.02 ^a^	77.00 ^b^	75.90 ^b^	72.49 ^ab^
SEM	2.01	2.28	1.88	1.71
*p*-value	<0.001	<0.001	<0.001	<0.001

Data are presented as means ± SEM. Numbers 2, 4, 6, and 8 represent the incubated times of 2 h, 4 h, 6 h, and 8 h, respectively. SEM, standard error of the mean; ^a–c^ means within columns with different superscript letters are different per *p* < 0.05.

**Table 6 biology-11-00044-t006:** Antibiotic resistances of the *Lactobacillus* strains.

Antioxidant Activity	*LAB* Strains							
	CY2	CY3	CY7	CY11	CY12	CY12-2	CY13	ATCC 7469
Tetracycline	S	S	S	S	S	S	I	S
Kanamycin	R	R	R	I	I	I	R	R
Penicillin	S	S	S	S	S	S	S	S
Gentamicin	R	I	I	R	I	I	R	I
Chloramphenicol	S	S	S	S	S	S	S	S
Ciprofloxacin	I	S	S	S	S	S	I	S
Oxacillin	R	R	R	R	R	R	R	R
Amoxicillin	S	S	R	S	S	S	S	S
Ampicillin	S	S	R	S	S	S	S	S
Erythrocin	S	S	S	S	S	S	S	S
Cefazolin	S	S	R	S	S	I	S	I
Vancomycin	R	R	R	R	R	R	R	R

S = sensitive (zone diameter ≥ 1.7 cm); I = intermediate (zone diameter from 1.2 to 1.7 cm); R = resistant (zone diameter ≤ 1.2 cm).

**Table 7 biology-11-00044-t007:** General genomic information of the strain CY12.

Indicator	CY12
Total reads num	601,513,839
Total bases	97,993
Average length	6138
No. of all scaffolds	1
Bases in all scaffolds	2,923,707
G + C content	46.77%
Gene number	2844
Gene total length(bp)	2,506,167
Gene average length(bp)	881
GC content in gene region (%)	47.35
Gene/Genome (%)	85.72
Number of coding sequences	1347
tRNA	59
rRNA	-

## Data Availability

The data presented in this study are available in the present study and Appendix A. The 16S sequences were deposited in the National Center for Biotechnology Information (NCBI) at GenBank with the accession numbers OK605898–OK605904; the whole-genome sequence read data were deposited in the National Center for Biotechnology Information (NCBI) at Sequence Read Archive (SRA) with the following accession number PRJNA770836 (https://dataview.ncbi.nlm.nih.gov/object/PRJNA770836) (13-OCT-2021).

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
