# Peer review of "In Vitro Probiotic Characteristics and Whole Genome Sequence Analysis of Lactobacillus Strains Isolated from Cattle-Yak Milk"

_biology, 2021, doi:10.3390/biology11010044_

Round 1
Reviewer 1 Report
The manuscript entitled “In Vitro Probiotic Characteristics and Whole Genome Sequence Analysis of Lactobacillus Strains Isolated from Cattle‐yak Milk” revealed that the Lactobacillus rhamnosus CY12 strain might be an excellent potential probiotic for the development of feed additives for animals to promote health by using a series of in vitro tests, and whole‐genome sequencing analysis. Overall, the work was well designed and executed and the presenting data can suitably support their conclusions. I have some points associated with this manuscript.
- The results of hemolytic activity teats were not shown in the manuscript.
- All selected strains were resistant to oxacillin and vancomycin, the author should explain more about why strain CY12 is still has the potential to develop as a probiotic.
Author Response
Responses to the comments of the Reviewer 1
Point 1: The results of hemolytic activity teats were not shown in the manuscript.
Response 1: We apologize for the incomplete data listed in Results. This should be possible now and has been amended as Figure S2 in Supplementary Materials.
Point 2: All selected strains were resistant to oxacillin and vancomycin, the author should explain more about why strain CY12 is still has the potential to develop as a probiotic.
Response 2: Thank you for your query. It is well-known that Lactobacillus strains are usually resistance toβ-lactam antibiotics such as oxacillin, ceftriaxone, and ampicillin, is due to the presence ofβ-lactamase in lactic acid bacteria(Jose et al., 2015, Reuben et al., 2020). Similarly, resistance to glycopeptides (vancomycin) has been noted in LAB, which is linked (in most cases) with innate resistance caused by membrane impermeability, probably via a resistance efflux mechanism (Liasi et al., 2009). However, our results showed that CY12 strain presented are susceptible to most antibiotics, that’s why CY12 strain still has the potential to develop as a probiotic.
Jose, N. M.; Bunt, C. R.; Hussain, A. M. Comparison of microbiological and probiotic characteristics of lactobacilli isolates from dairy food products and animal rumen contents. Microorganisms.2015, 3,198–212
Reuben, R.C.; Roy, P.C.; Sarkar, S.L.; Rubayet Ul Alam, A.S.M.; Jahid, I.K. Characterization and evaluation of lactic acid bacteria from indigenous raw milk for potential probiotic properties. J Dairy Sci. 2020, 103, 1223-37.
Liasi, S. A.; Azmi, T. I.; Hassan, M. D.; Shuhaimi, M.; Rosfarizan, M.; Ariff, A. B. Antimicrobial activity and antibiotic sensitivity of three isolates of lactic acid bacteria from fermented fish product, Budu. Malays. J. Microbiol. 2009, 5, 33-37.
Reviewer 2 Report
- Having simple summary and abstract is redundant. So if not required by format, simple summary can be removed. Abstract in current form in acceptable. But the words, methods, results, conclusion can be removed form lines 27, 29 and 37 respectively.
- Line 46-48, reword this sentence, too much information so split into two or three sentences. Use verb in correct format.
- Line 84, ….study was to evaluate and to identify…..
- Line 145, 147, kindly mention source of pepsin and trypsin respectively
- Line 190, how many bacteria were put into each well. Kindly mention infectivity ratio of bacteria to cells. This is critical, as number of bacteria adhering to cells, would directly depend on number of bacteria with respect to cells. This ratio has to be kept in all wells. Was experiment performed using the same ratio in each well? If not, adhesion experiment needs to be repeated with same ration in all wells.
- Line 192, wished need to be replaced with washed
Author Response
Responses to the comments of the Reviewer 2
Point 1: Having simple summary and abstract is redundant. So if not required by format, simple summary can be removed. Abstract in current form in acceptable. But the words, methods, results, conclusion can be removed form lines 27, 29 and 37 respectively.
Response 1: Thanks for your suggestions, the words such as methods, results, and conclusion have deleted. However, simple summary is required by format, that’s why we did not delete it in the manuscript. Please apologize. We now put much emphasis on the style and format rules. In some minor issues the format of recently published articles differed from the format of the sample manuscript. Here we finally decided to follow the style of the recently published (“Articles in Press”) article format.
Point 2: Line 46-48, reword this sentence, too much information so split into two or three sentences. Use verb in correct format.
Response 2: We have rewritten this part. See for details: “there is a common tendency to reduce or avoid the use of antibiotics in animal husbandry. Hence, it is crucial to find an ideal substitute for antibiotics. Nowadays, Prebiotics as a second-generation substitute for antibiotics has been the hotspot in the research on animal nutrition.”
Point 3: Line 84, ….study was to evaluate and to identify…..
Response 3: Corrected as suggested.
Point 4: Line 145, 147, kindly mention source of pepsin and trypsin respectively
Response 4: The reviewer is correct. Now gave as “Solarbio, Beijing, China” and should be possible in page 4.
Point 5: Line 190, how many bacteria were put into each well. Kindly mention infectivity ratio of bacteria to cells. This is critical, as number of bacteria adhering to cells, would directly depend on number of bacteria with respect to cells. This ratio has to be kept in all wells. Was experiment performed using the same ratio in each well? If not, adhesion experiment needs to be repeated with same ration in all wells.
Response 5: Thank you for your question. The selected strains were cultured in MRS broth (3%v/v) at 37 ℃ for overnight before experiment. And bacteria added to each well at a concentration of 108 CFU/mL and co-cultured at 37 ℃ for 3 h. And detailed description can be found in page 5.
Point 6: Line 192, wished need to be replaced with washed
Response 6: We have made the correction.
Reviewer 3 Report
The aim of this study was to identify the beneficial strains from the milk of
cattle‐yak, and evaluate probiotic characteristics of potential strains through using a series of in vitro tests. The manuscript is interesting, however, there are some comments that need to be addressed
Major concerns
1- The manuscript needs English proofreading by a native English speaker including for the title.
Other concerns
1- Write (in vitro) in italic throughout the manuscript.
2- Please add references to the methodology section 2.2.1 and 2.2.2.
3- Did you check the invasion and intracellular survival in Caco‐2
4- Figure 1: write in the legends, what is the control in both figures.
5- Name of the pathogens or any bacteria need to be full in the first time you write in the manuscript then write the abbreviation in the rest of the manuscript. for example E. coli
6- Table 1 and 2 and 3 and 4 and 5: write in the footnote, what are these numbers in the table, they represent what?
7- Correct the first raw of table 5 and put the time (h) in the right place
8- You need to add rationale in materials and methods. why you made WGS only for strain CY12 strain?
Author Response
Responses to the comments of the Reviewer 3
Major concerns
Point 1: The manuscript needs English proofreading by a native English speaker including for the title.
Response 1: We apologize for the language problems in the original manuscript. The language presentation was improved with assistance from a native English-speaking Scientists with appropriate research background.
Other concerns
Point 1: Write (in vitro) in italic throughout the manuscript.
Response 1: Corrected.
Point 2: Please add references to the methodology section 2.2.1 and 2.2.2.
Response 2: The reviewer is right. All serial numbers have now been checked and amended where necessary.
Point 3: Did you check the invasion and intracellular survival in Caco‐2
Response 3: We understand your point and will keep it in mind during our future experiments. We have added the limitations of our used method in the discussion. Antimicrobial activity is one of the important criteria for selecting novel potential probiotic strains. Meanwhile, the adhesion to the mucosal surfaces and epithelial cells is considered as a crucial characteristic for candidate probiotics selection, and play important roles in competitively excluding or inhibiting pathogens. Therefore, in this study, we mostly studied the antimicrobial activity and adherence ability of the selected LAB. However, the selected strains whether could inhibit invasion and intracellular survival of the bacteria needed to further researched.
Point 4: Figure 1: write in the legends, what is the control in both figures.
Response 4: We apologize for this. It is corrected now.
Point 5: Name of the pathogens or any bacteria need to be full in the first time you write in the manuscript then write the abbreviation in the rest of the manuscript. for example E. coli
Response 5: We have made the changes.
Point 6: Table 1 and 2 and 3 and 4 and 5: write in the footnote, what are these numbers in the table, they represent what?
Response 6: Done it as your suggestion.
Point 7: Correct the first raw of table 5 and put the time (h) in the right place
Response 7: Corrected as suggestion.
Point 8: You need to add rationale in materials and methods, why you made WGS only for strain CY12 strain?
Response 8: Thank you for this statement on novelty and the other statements. Based on your comment, we have explained why we made WGS only for CY12 strain. And relative description added in the results and discussion. see for details: “Overall, compared to other selected strains, the CY12 strains showed a high survival rate in bile salts, acid condition, and gastrointestinal juice environment, as well as showed high antimicrobial activity and adhesive potential. Hence, CY12 strains was selected as the next experimental research object to further explore the potential biological functions and information.”
“In this study, the CY12 strain was selected according to probiotic features and safety evaluation. Genome-wide sequencing was performed on L. rhamnosus CY12 strain, to further better understand their potential biological functions.”
Round 2
Reviewer 2 Report
Thanks for reply and making required changes.
Reviewer 3 Report
The reviewers have addressed all my concerns